# NON-SYN VARIATIONAL AUTOENCODERS

## ABSTRACT

Learning disentangling representations of the independent factors of variations that explain the data in an unsupervised setting is still a major challenge. In the following paper we address the task of disentanglement and introduce a new state-of-the-art approach called Non-synergistic variational Autoencoder (Non-Syn VAE). Our model draws inspiration from population coding, where the notion of synergy arises when we describe the encoded information by neurons in the form of responses from the stimuli. If those responses convey more information together than separate as independent sources of encoding information,they are acting synergetically. By penalizing the synergistic mutual information within the latents we encourage information independence and by doing that disentangle the latent factors. Notably, our approach could be added to the VAE framework easily, where the new ELBO function is still a lower bound on the log likelihood $p_x$. In addition, we qualitatively compare our model with Factor VAE and show that this one implicitly minimises the synergy of the latents.

## 1 INTRODUCTION

Our world is hierarchical and compositional, humans can generalise better since we use primitive concepts that allow us to create complex representations (Higgins et al. (2016)). Towards the creation of truly intelligent systems, they should learn in a similar way resulting in an increase of their performance since they would capture the underlying factors of variation of the data ( Bengio et al. (2013); Hassabis et al. (2017); Botvinick et al. (2017)). In addition, good representations improve the performance for tasks involving transfer learning and multi-task learning; since it will capture the explanatory factors.

According to Lake et al. (2016), a compositional representation should create new elements from the combination of primitive concepts resulting in a infinite number of new representations. For example if our model is trained with images of white wall and then is presented a boy with a white shirt, it should identify the color white as a primitive element. Intuitively, our model will be able to construct different and multiple representations from the primitives.

Furthermore, a disentangled representation has been interpreted in different ways, for instance Bengio et al. (2013) define it as one where single latent variables are sensitive to changes in generative factors, while being invariant to changes in other factors. In addition, we agree with Higgins et al. (2017a), which mentions that a disentangle representation should be factorised and interpretable. Intuitevely, the model could learn generative factors such as position, scale or colour; if it is disentangle it should be able to traverse along the position variable without changing the scale or the colour. It's worth noting that disentangled representations have been useful for a variety of downstream tasks such as domain adaptation by training a Reinforcement Learning agent that uses a disentangled representation of its environment Higgins et al. (2017b); or for learning disentangled primitives grounded in the visual domain discovered in an unsupervised manner Higgins et al. (2017c).

## 2 RELATED WORK

The original Variational autoencoder (VAE) framework (Kingma & Welling (2013); Rezende et al. (2014)) has been used extensively for the task of disentanglement by modifying the original ELBO formulation; for instance $\beta$-VAE is presented in Higgins et al. (2017a) which increases the latent capacity by penalising the KL divergence term with a $\beta$ hyperparameter. In addition, Kim & Mnih

(2018) achieved a more robust disentangled representation by using the model called Factor VAE which penalises the total correlation of the latent variables encouraging the independence of the latents; a similar approach is shown in Chen et al. (2018), where they present a clever ELBO decomposition based on Hoffman & Johnson (2016). Other approaches rely on information bottleneck presented in Tishby et al. (1999) to model frameworks for this task such as Alemi et al. (2016); Achille & Soatto (2016). Furthermore, Chen et al. (2016) describe a model based on Generative Adversarial Networks (Goodfellow et al. (2014)) by encouraging the mutual information between the latents and the output of the generator. Notably, in Higgins et al. (2016), they describe comprehensively the $\beta$-VAE model using a neuroscience and information theory approaches; they suggest that by encouraging redundancy reduction the model achieves statistical independence within the latents. This model inspired us to look into different fields for new ways to enforce disentanglement of the latents.

## 3 SYNERGY

To understand our model, we need first to describe Synergy (Gat & Tishby (1998); Schneidman et al. (2003)) being a popular notion of it as how much the whole is greater than the sum of its parts. It's common to describe it with the XOR gate, since we need two independent variables to fully specified the value of the output. Following, we describe the synergy from two related fields.

### 3.1 INFORMATION THEORY APPROACH

Computing the multivariate synergistic information is an ongoing topic of research Schneidman et al. (2003); Williams & Beer (2010); Bertschinger et al. (2012); Griffith & Koch (2012). Most of the current research in this topic uses the Partial information diagram described by Williams & Beer (2010). In order to understand the importance of the Synergy information in our framework it's essential to describe the relations with the Unique and Redundant Information. Introducing the notation from Williams & Beer (2010), let's consider the random variable S and a random vector $\boldsymbol{R} = \{R_1, R_2, .., R_n\}$, being our goal to decompose the information that the variable $R$ provides about S; the contribution of these partial information could come from one element from $R_1$ or from subsets of $R$ (ie. $R_1, R_2$). Considering the case with two variables, $\{R_1, R_2\}$, we could separate the partial mutual information in unique ($Unq(S; R_1)$ and $Unq(S; R_2)$ ), the information that only $R_1$ or $R_2$ provides about S is redundant ($Rdn(S; R_1, R_2)$), which it could be provided by $R_1$ or $R_2$; and synergistic ($Syn(S; R_1, R_2)$), which is only provided by the combination of $R_1$ and $R_2$. The figure 1 depicts the decomposition; this diagram is also called PI-diagram (Partial information) . Additional notation is used for a better visualisation in the case of more variables: Unique $\{1\},\{2\}$; Redundant $\{1\}\{2\}$ and Synergistic $\{12\}$.

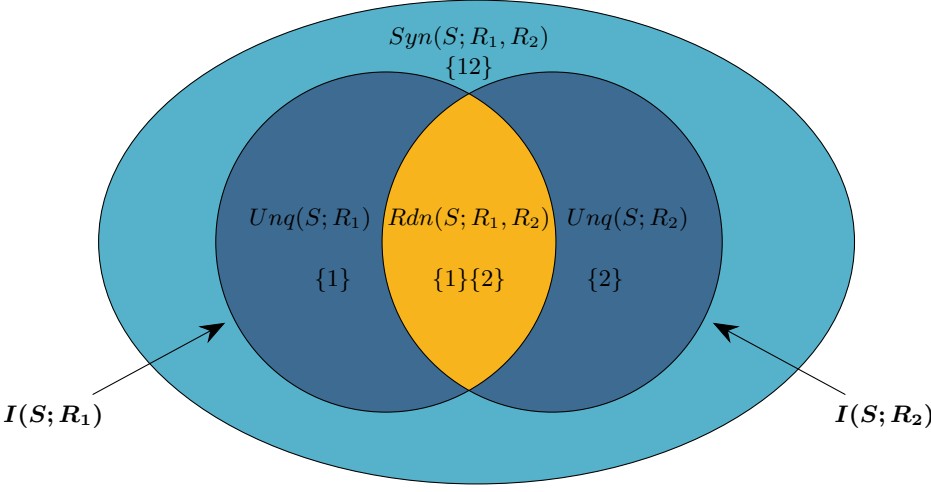

Figure 1: Structure of total information of two variables about S

For the case of 2 variables $(X_1, X_2)$, we expect four contributions to the mutual information as described in Bertschinger et al. (2012); Olbrich et al. (2015):

$$I(S; R_1, R_2) = \underbrace{SI(S; R_1, R_2)}_{\text{Redundant}} + \underbrace{Unq(S; R_1 \setminus R_2)}_{\text{Unique}} + \underbrace{Unq(S; R_2 \setminus R_1)}_{\text{Unique}} + \underbrace{Syn(S; R_1, R_2)}_{\text{Synergistic}} \quad (1)$$

It's easy to see that the number of terms increases exponentially as the number of sources increases. The best measure for synergy is an ongoing topic of research. In the subsection we are going to talk about the synergy metrics.

## 3.2 POPULATION CODING APPROACH

For neural codes there are three types of independence when it comes to the relation between stimuli and responses; which are the activity independence, the conditional independence and the information independence. One of the first measures of synergy for sets of sources of information came from this notion of independence. In Williams & Beer (2010) it is stated that if the responses come from different features of the stimulus, the information encoded in those responses should be added to estimate the mutual information they provide about the stimulus. Formally:

$$I(S; R_1, R_2) = I(S; R_1) + I(S; R_2) \quad (2)$$

However, we just saw in the previous sections that the $I(S; R_1)$ and $I(S; R_2)$ could be decomposed in their unique and redundant and synergistic terms. Intuitively, this formulation only holds if there is no redundant or synergistic information present; which means in the context of population coding that the responses encoded different parts of the stimulus. If the responses $R_1$ and $R_2$ convey more information together than separate, we can say we have synergistic information; if the information is less, we have redundant information. That's the reason why in Gat & Tishby (1998), the synergy is considered as measure of information independence:

$$Syn(R_1, R_2) = I(S; R_1, R_2) - I(S; R_1) - I(S; R_2) \quad (3)$$

## 3.3 SYNERGY METRIC

First we need to state the notation (the same as Griffith & Koch (2012)):

- n: Number of individual predictors $X_i$
- $\mathbb{A}_i$ : subset of individual predictors (ie. $A_i = \{X_1, X_3\}$)
- **X**: Joint random variable of all individual predictors $X_1 X_2 .. X_n$
- $\{X_1, X_2, ..., X_n\}$: Set of all the individual predictors
- Y: Random variable to be predicted
- y: A particular outcome of Y.

The intuition behind this metric is that synergy should be defined as the "whole beyond the maximum of its parts". The whole is described as the mutual information between the joint **X** and the outcome Y; whereas the maximum of all the possible subsets is interpreted as the maximum information that any of the sources $\mathbb{A}_i$ provided about each outcome. Formally, this is stated as:

$$S_{max}(\{X_1, X_2, ..., X_n\}; Y) = I(\boldsymbol{X}; Y) - I_{max}(\{\mathbb{A}_1, \mathbb{A}_2 .. \mathbb{A}_n\}; Y) \quad (4)$$

$$= I(\boldsymbol{X}; Y) - \sum_{y \in Y} p(Y = y) \max_i I(\mathbb{A}_i; Y = y) \quad (5)$$

This metric derives from Williams & Beer (2010) and Griffith & Koch (2012), however one of the differences with this metric with the one presented in Griffith & Koch (2012) is that in this one we

are considering the specific mutual information $I_{max}$ in a group of latents $\mathbb{A}_i$, whereas in the paper mentioned it considers only an individual latent. Notably the $I_{max}$ can be expressed in terms of the KL divergence.

$$I(\mathbb{A}_i; Y = y) = \sum_{a_i \in \mathbb{A}i} P(a_i \mid y) \log \frac{P(a_i, y)}{P(a_i)P(y)} \tag{6}$$

$$= D_{KL}\big[P(\mathbb{A}_i \mid y) \parallel P(\mathbb{A}_i)\big] \tag{7}$$

Putting together the equation 7 and 5, we have the following:

$$S_{max}(\{X_1, X_2, ..., X_n\}; Y) = I(\boldsymbol{X}; Y) - \sum_{y \in Y} p(Y = y) \max_i D_{KL}\big[P(\mathbb{A}_i \mid y) \parallel P(\mathbb{A}_i)\big] \tag{8}$$

In the following section we are going to use the intuition provided by the above equation in the VAE framework for the task of disentanglement.

## 4 MODEL DERIVATION

The motivation of our contribution is inspired in this concept and driven by the belief that synergy is not desirable for the task of disentanglement, since we want the latents to be independently informative as possible about the data, instead of needing many latents to specify the data. Therefore, we argue that by penalising the synergistic information within the latents and the data, we would encourage the disentanglement of the underlying factors of variation. This hypothesis is also inspired in the information presented in Griffith & Koch (2012); Gat & Tishby (1998), where it is stated that the synergy is a measure of independence information in the responses of the stimuli.

First, we need to change the notation to match the VAE framework notation ($Z$ are the latents and $X$ is the observations). Also, $\mathbb{A}_i$ is a subset of the latents, such that $A_i \in \{Z_1, Z_2, ..., Z_n\}$ and $\boldsymbol{Z}$ is the joint of the latents. Formally: $\boldsymbol{Z} = \prod_i^d Z_i$, where $d$ is the number of dimensions of the latent variables. Besides, from the VAE standard framework, we know that the distribution $p(\mathbb{A}_i|x)$ is intractable which is why we need to use an approximate distribution $q_\phi(\mathbb{A}_i|x)$ parametrised by $\phi$ parameters. It's important to notice that this KL divergence could be computed in the same way as in the VAE framework; the only difference is the number of dimensions used for the random variable $z$. In the original VAE framework, we compute the KL divergence considering the joint $\boldsymbol{Z} = \prod_i^d Z_i$; whereas for the Synergy metric we don't use the joint but a subset of the latents. For instance, if $A_i = Z_2 Z_5 Z_8$, we have the following expression:

$$D_{KL}\big[q_\phi(z_2 z_5 z_8 \mid x) \parallel p(z_2 z_5 z_8)\big] \tag{9}$$

Taking in account these considerations, we express the equation 8 as follows:

$$S_{max}(\{Z_1, Z_2, ..., Z_d\}; X) = I(\boldsymbol{Z}; X) - \sum_{x \in X} p(X = x) \max_i D_{KL}\big[q_\phi(\mathbb{A}_i \mid x) \parallel p(\mathbb{A}_i)\big] \tag{10}$$

We start with the original ELBO formulation (Rezende et al. (2014); Kingma & Welling (2013)) and add the penalised term corresponding to the synergy, where $\alpha$ is a hyperparameter:

$$\mathcal{L}_{elbo}(\theta, \phi, x) = \mathbb{E}_{q_\phi(z|x)}\big[\ \log p_\theta(x \mid z)\big]\ - D_{KL}\big[q_\phi(z \mid x) \parallel p(z)\big] \tag{11}$$

$$\mathcal{L}_{new}(\theta, \phi, x) = \mathcal{L}_{elbo}(\theta, \phi, x) - \alpha S_{max}(\{Z_1, Z_2, ..., Z_d\}; X) \tag{12}$$

Expanding the $S_{max}$ term, we have the llowing:

$$\mathcal{L}_{new}(\theta, \phi, x) = \mathcal{L}_{elbo}(\theta, \phi, x) - \alpha(I(z; x) - \sum_{x \in X} p(X = x) \max_i D_{KL}\big[q_\phi(A_i | x) \parallel p(A_i)\big]) \quad (13)$$

From Hoffman & Johnson (2016), we know that the KL term in the ELBO loss is decomposed in $D_{KL}\big[q_\phi(z_n) \parallel p(z_n)\big] + I(x_n; z)$ when we use the aggregate posterior and define the loss over the empirical distribution of the data $p_{data}(x)$. Taking in account that, we can express the equation 12 as follows:

$$\mathcal{L}_{new}(\theta, \phi, x) = \frac{1}{N} \sum_{i=1}^{N} \left[ \mathbb{E}_{q_\phi(z|x)}\big[ \log p_\theta(x^{(i)} | z)\big] \right] - D_{KL}\big[q_\phi(z_n) \parallel p(z_n)\big] - I(x_n; z)$$

$$\underbrace{-\alpha I(x_n; z)}_{\text{Penalise}} + \alpha \sum_{x \in X} p(X = x) \max_i D_{KL}\big[q_\phi(\mathbb{A}_i | x) \parallel p(\mathbb{A}_i)\big]) \quad (14)$$

If we penalise the synergy (see Eq. 12), we will be penalising the mutual information term which is not desirable for this task Kim & Mnih (2018); we can see this effect explicitly in Eq. 14. Therefore, we use only the second term to perform the optimisation which means maximising the subset of latents with the most amount of MI per outcome.

$$\mathcal{L}_{new}(\theta, \phi, x) = \mathcal{L}_{elbo}(\theta, \phi, x) + \alpha \sum_{x \in X} p(X = x) \max_i D_{KL}\big[q_\phi(\mathbb{A}_i | x) \parallel p(\mathbb{A}_i)\big]) \quad (15)$$

It's easy to see in Eq. 15 that it's not a guaranteed lower bound on the log likelihood $p_x$ anymore, which is why we decided to penalise the subset of latents with the minimum specific mutual information (ie. $\mathbb{A}_w$). In practice we found that computing the maximum subset $\mathbb{A}_i$ for each outcome of x is too computational intensive, which is why we decided to use a mini-batch approximation, which is the version we show in the pseudo-code using a two step optimisation in the next section. The final version we are going to use is the one below, where Imax is the KL term of the synergy term.

$$\mathcal{L}_{new}(\theta, \phi, x) = \underbrace{\mathbb{E}_{q_\phi(z|x)}\big[ \log p_\theta(x|z)\big] - D_{\text{KL}}(q_\phi(z|x) \parallel p(z))}_{\mathcal{L}_{elbo}} - \underbrace{\alpha D_{KL}\big[q_\phi(\mathbb{A}_w|x) \parallel p(\mathbb{A}_w)\big]}_{\alpha * \text{Imax}}$$

$$(16)$$

---

**Algorithm 1** Non Syn VAE

---

**Input:** Observations $(x^{(i)})_{i=1}^{N}$, batch size $m$, latent dimension $d$, weight of synergy loss $\alpha$, discount factor $\omega$, optimiser $optim$, reparametrisation function $g_\phi$.

> $\theta, \phi \leftarrow$ Initialise VAE parameters
> **repeat**
> 3:     $x^{(i)} \leftarrow$ Random minibatch B of size m, $i \in B$
>         $z_i \leftarrow g_\phi(\epsilon, x^{(i)})$                                              ▷ Sample $z_i \sim q_\phi(z|x)$
> 6:     $\phi, \theta \leftarrow \text{optim}(\nabla_{\theta, \phi} \mathcal{L}_{elbo}(\theta, \phi; x))$         ▷ : Gradients of ELBO minibatch
>         $x'^{(i)} \leftarrow$ Random minibatch B' of size m, $i \in B'$
>         $mu, logvar \leftarrow \text{Encoder}(x'^{(i)}, \phi)$
> 9:     $worst\_index \leftarrow \textbf{get\_index\_greedy}(mu, logvar, \omega)$
>         $\mathcal{L}_{syn} \leftarrow \alpha * \textbf{Imax}(mu, logvar, worst\_index)$         ▷ See Eq.16 for Imax function
>         $\phi \leftarrow \text{optim}(\nabla_\phi \mathcal{L}_{syn}(\phi; x'^{(i)}))$         ▷ : Gradients of Syn loss minibatch
> 12: **until** convergence of objective

---

In the algorithm shown above we see in practice that we get better results when we sample the values of $mu$ and $logvar$ from the encoder for the step 2 of the optimisation. We use a greedy approximation of the best latents by following a greedy policy (See Appendix).

## 5 EXPERIMENTS

### 5.1 LATENT TRAVERSALS

For disentanglement, the dataset most commonly used is the dsprites dataset Matthey et al. (2017), which consists on 2D shapes generated from independent latent factors. We used the same architecture and optimizer as Factor VAE Kim & Mnih (2018) for training our model. In order to test qualitatively the Non-Syn VAE model, we decided to tranverse the latents and plot the mean activations of the latents.

In Figure 2 (left), we see clearly that our model disentangles the factors of variation. Likewise, on the right we see the mean activation of each active latent averaged across shapes, rotations and scales.

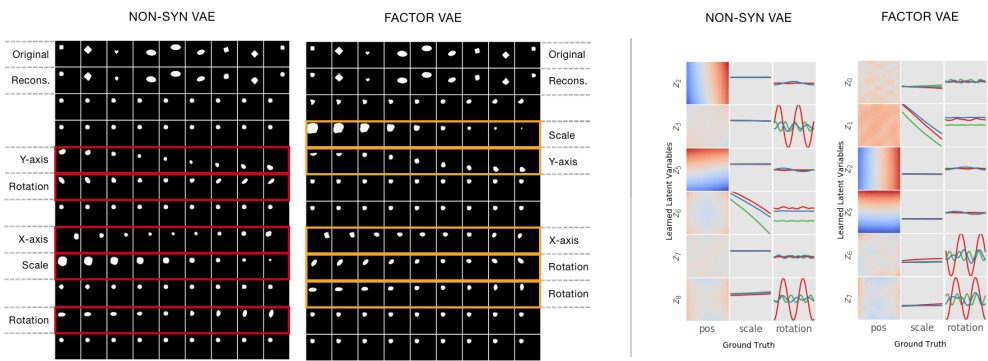

Figure 2: Left: Traverse of latents (110k steps). Right: Mean activations (110k steps)

After looking at the figure above we can state that our model achieves state-of-the-art results using a qualitatively benchmark. Interestingly, both models perform quite similar in this test.

### 5.2 SYNERGY IN FACTOR VAE

Also, we decided to compute the same synergy term from the Non-Syn VAE in Factor VAE (just compute it, we didn't use it for training). The hypothesis was that if Factor VAE achieves disentanglement, it should minimise the synergy as well. We train Factor VAE using the same parameters and architecture described in Kim & Mnih (2018). We show the first 4000 steps for the Synergy term (i.e. $\alpha * Imax$), since most of the interaction happens in the first steps.

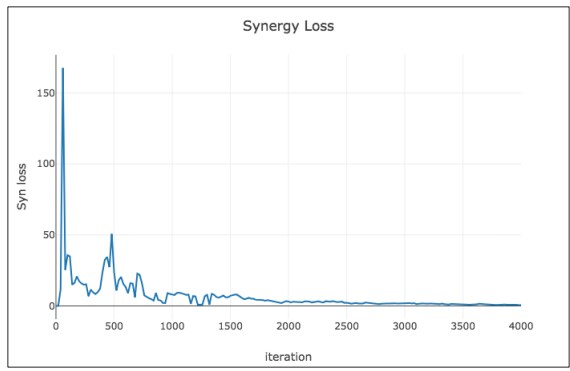

Figure 3: Synergy loss for Factor VAE - 4k steps

As a comparison, we also show the synergy for the Non-Syn VAE for the same number of steps in Fig 4. Surprisingly, Factor VAE minimises the Synergy implicitly by penalising the Total correlation term.

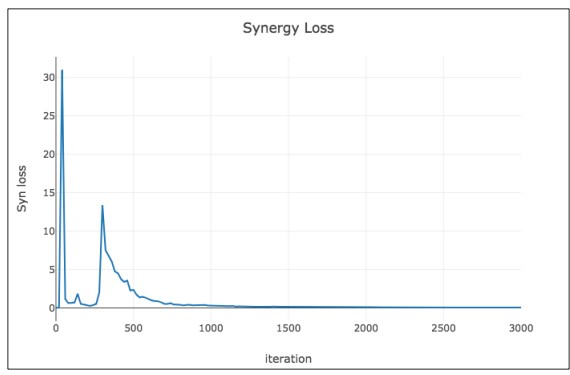

Figure 4: Synergy loss for Non-Syn VAE - 4k steps

## 6  CONCLUSIONS

In this paper we presented the intuition and derivation of the lower bound of a model that uses a novel approach inspired by the information theory and Neuroscience fields to achieve the disentanglement of the underlying factor of variations in the data. After looking at the results,we can state that our model achieved state-of-the-art results, with a performance close to FactorVAE Kim & Mnih (2018).

This is not the first time that a model draws ideas from information theory. Many models Tishby et al. (1999); Alemi et al. (2016); Achille & Soatto (2016) used the information bottleneck presented in Tishby et al. (1999) using the VAE framework. Therefore, we truly believe that we should keep looking at the neuroscience and information theory fields for inspiration. In general, we don't need to replicate or simulate biological models; however we should analyse the intuition about the known main mechanisms of our brain and adapt those to our models.

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

# 7 APPENDIX

## 7.1 DETAILS OF TRAINING

We trained the Non-Syn VAE model for 150,000 steps using the d-sprites data set (unsupervised setting) obtaining the best result using the following parameters:

- Optimiser: Adam
- Learning rate: 1e-4
- Beta 1 (Adam): 0.9
- Beta 2 (Adam): 0.999
- Batch size: 64
- Discount $\omega$ of greedy policy: 0.9
- Weight of the Synergy loss $\alpha$: 5.0

## 7.2 ARCHITECTURE OF NON-SYN VAE - DSPRITES

| Encoder | Decoder |
|---------|---------|
| Input 64 x 64 binary image | Input |
| 4 x 4 conv.32 ReLU, stride 2 | FC. 128 ReLU |
| 4 x 4 conv.32 ReLU, stride 2 | FC. 4 x 4 x 64 ReLU. |
| 4 x 4 conv.64 ReLU, stride 2 | 4 x 4 upconv.64 ReLU, stride 2 |
| 4 x 4 conv.64 ReLU, stride 2 | 4 x 4 upconv.32 ReLU, stride 2 |
| FC. 128 | 4 x 4 upconv.32 ReLU, stride 2 |
| FC. 2x10 | 4 x 4 upconv.1, stride 2 |

## 7.3 DETAILS OF ALGORITHM

---
**Algorithm 2** Imax

**Input:** Compute specific mutual information
**Output:**

1: **function** IMAX($mu, logvar, index$)
2:     $mu\_syn \leftarrow mu[:, index]$
3:     $logvar\_syn \leftarrow logvar[:, index]$
4:
5:     Imax $\leftarrow$ **compute_KL**($mu\_syn, logvar\_syn$)
6:
7:     **return** $Imax$
---

---

**Algorithm 3** generate_candidate

---

**Input:** dimension of Z, index
**Output:** List of indices

1: **function** GENERATE_CANDIDATE($d, index$)
2:     **if** $len(d) = 0$ **then**
3:         $candidates \leftarrow [1:10]$
4:     **else**
5:         $candidates \leftarrow [1:10]$ not in $index$
6:     **return** $candidates$

---

**Algorithm 4** get_index_greedy

---

**Input:** mu $\mu$, log std. deviation $\sigma$, discount factor $\omega$
**Output:** list of indices of the latents with the lowest Specific Mutual Information

    **function** GET_INDEX_GREEDY($mu, logvar, \omega$)
2:     **for** $i \leftarrow 1$ to $d$ **do**
        $candidates \leftarrow$ **generate_candidate**($d, index$)         $\triangleright \oplus$: bitwise exclusive-or
4:

        **for** $c \leftarrow 1$ in $candidates$ **do**
6:             index $\leftarrow best\_index$ + [c]
            $Imax\_new \leftarrow$ **Imax**($\mu, \sigma, index$)
8:

            **if** $Imax\_new * \omega > I\_max\_best$ **then**
10:                $Imax\_best \leftarrow Imax\_new$
               $best\_index \leftarrow index$
12:
      $worst\_index \leftarrow [1:10]$ not in $best\_index$
14:     **return** $worst\_index$

---

