# OpenReview forum: "Non-Synergistic Variational Autoencoders"
_ICLR.cc/2019/Conference_

### Official Review · AnonReviewer3 · 2018-10-14
**Far off being ready for publication**

**Rating:** 3
**Confidence:** 5

**Review:**

This paper proposes a new approach to enforcing disentanglement in VAEs using a term that penalizes the synergistic mutual information between the latent variables, encouraging representations where any given piece of information about a datapoint can be garnered from a single latent.  In other words, representations where there is no information conveyed by combinations of latents that is not conveyed by considering each latent in isolation.  As the resultant target is intractable to evaluate, a number of approximations are employed for practical training.

The high-level idea is quite interesting, but the paper itself is quite a long way of convincing me that this is actually a good approach.  Moreover, the paper is a long way of the level of completeness, rigor, clarity, and polish that is required to seriously consider it for publication.  In short, the work is still at a relatively early stage and a lot more would need to be done for it to attain various minimum standards for acceptance.  A non-exhaustive list of specific examples of its shortfalls are given below.

1. The paper is over a page and a half under length, despite wasting large amounts of space (e.g. figures 3 and 4 should be two lines on the same plot)

2. The experimental evaluation is woefully inadequate.  The only quantitative assessment is to compare to a single different approach on a single toy dataset, and even then the metric being used is the one the new method uses to train for making it somewhat meaningless.

3. The introduction is completely generic and says nothing about the method itself, just providing a (not especially compelling) motivation for disentanglement in general.  In fact, the motivation of the introduction is somewhat at odds with the work -- correctly talking about the need for hierarchical representations which the approach actually actively discourages.

4. There are insufficient details on the algorithm itself in terms of the approximations that are made to estimate the synergistic mutual information.  These are mostly glossed over with only a very short explanation in the paragraph after equation 15.  Yes there are algorithm blocks, but these are pretty incomprehensible and lack accompanying text.  In particular, I cannot understand what A_w is supposed to be.  This is very important as I suspect the behavior of the approximation is very different to the true target.  Similarly, it would be good to provide more insight into the desired target (i.e. Eq 15).  For example, I suspect that it will encourage a mismatch between the aggregate posterior and prior by encouraging higher entropy on the former, in turn causing samples from the generative model to provide a poor match to the data.

5. The repeated claims of the approach and results being "state-of-the-art" are cringe-worthy bordering on amusing.  Writing like this serves no purpose even when it justified, and it certainly is not here.

6. There are a lot of typos throughout and the production values are rather poor.  For example, the algorithm blocks which are extremely messy to the point where they are difficult to follow, citep/citet mistakes occur almost every other citation, there is a sign error in Equation 16.


This is a piece of work in an exciting research area that,  with substantial extra work, could potentially result in a decent paper due to fact that the core idea is simple and original.  However, it is a long way short of this in its current state.  Along with addressing the specific issues above and improving the clarity of the work more generally, one thing in particular that would need to address in a resubmission is a more careful motivation for the method (ideally in the form of a proper introduction).

Though I appreciate this is a somewhat subjective opinion, for me, penalizing the synergistic information is probably actually a bad thing to do when taking a more long-term view on disentanglement.  Forcing simplistic representations where no information is conveyed through the composition of latents beyond that they provide in isolation is all well and good for highly artificial and simplistic datasets like dsprites, but is clearly not a generalizable approach for larger datasets where no such simplistic representation exists.  As you say in the first line of your own introduction, hierarchy and composition are key parts of learning effective and interpretable representations and this is exactly what you are discouraging.  A lot of the issue here is one of the disentanglement literature at large rather than this paper (though I do find it to be a particularly egregious offender) and it is fine to have different opinions.  However, it is necessary to at least make a sensible case for why your approach is actually useful.

Namely, is there actually any real applications where such a simplistic disentanglement is actually useful?  Is there are anyway the current works helps in the longer vision of achieving interpretable representations?  When and why is the synergistic information a better regularizer than, for example, the total correlation?  The experiments you have do not make any inroads to answering these questions and there are no written arguments of note to address them.  I am not trying to argue here that there isn't a good case to be made for the suggested approach in the context of these questions (though I am suspicious), just that if the work is going to have any lasting impact on the community then it needs to at least consider them.

---

### Official Review · AnonReviewer2 · 2018-10-31
**Important topic, but lack of experiments**

**Rating:** 4
**Confidence:** 3

**Review:**

The authors aim at training a VAE that has disentangled latent representations in a "synergistically" maximal way.
For this they  use one (of several possible) versions of synergy defintions and create a straight forward penalization term for a VAE objective (roughly the whole mutual information minus the maximum mutual information of its parts).
They train this VAE on one dataset, namely dsprites, and compare it to a VAE with total correlation penalization.

The paper is well written and readable. The idea of using synergy is an important step forward in understanding complex models. The concept of synergy has great potential in machine learning and is highly relevant.

The main concepts of synergy are not developed in this paper and the used penalization term is straight forward.
The number of experiments conducted and comparisons done is quite limited. Also the potential of synergy is not really demonstrated, e.g. for representation learning, causality, etc., and appears here ad hoc.
Also why one should use the authors' suggested penalization term instead of total correlation is not discussed, nor demonstrated as they perform similarly on both disentanglement and synergy loss.

I hope the authors find more relevant applications or data sets in the future to demonstrate the importance of synergy.

---

### Official Review · AnonReviewer1 · 2018-11-05
**Interesting perspective, but not strong enough results**

**Rating:** 3
**Confidence:** 4

**Review:**

The paper proposes a new objective function for learning disentangled representations in a variational framework, building on the beta-VAE work by Higgins et al, 2017. The approach attempts to minimise the synergy of the information provided by the independent latent dimensions of the model. Unfortunately, the authors do not properly evaluate their newly proposed Non-Syn VAE, only providing a single experiment on a toy dataset and no quantitative metric results. Furthermore, even qualitatively the proposed model is shown to perform no better than the existing factor-VAE baseline.

I commend the authors for taking a multi-disciplinary perspective and bringing the information synergy ideas to the area of unsupervised disentangled representation learning. However, the resulting Non-Syn VAE objective function is effectively a different derivation of the original beta-VAE objective. If the authors want to continue with the synergy minimisation approach, I would recommend that they attempt to use it as a novel interpretation of the existing disentangling techniques, and maybe try to develop a more robust disentanglement metric by following this line of reasoning. Unfortunately, in the current form the paper is not suitable for publication.

---

### Meta-Review · Area_Chair1 · 2018-12-18
**Interesting approach, but more work needed on theory and experiments**

**Confidence:** 4
**Recommendation:** Reject

**Metareview:**

The paper introduces a form of variational auto encoder for learning disentangled representations. The idea is to penalise synergistic mutual information. The introduction of concepts from synergy to the community is appreciated.

Although the approach appears interesting and forward looking in understanding complex models, at this point the paper does not convince on the theoretical nor on the experimental side. The main concepts used in the paper are developed elsewhere, the potential value of synergy is not properly examined.

The reviewers agree on a not so positive view on this paper, with ratings either ok, but not good enough, or clear rejection. There is a consensus that the paper needs more work.